# Cryo-electron microscopy structures of capsids and in situ portals of DNA-devoid capsids of human cytomegalovirus

Zhihai Li[1,2,3,6], Jingjing Pang[2,3,4,6], Rongchao Gao[2], Qingxia Wang[2], Maoyan Zhang[5] & Xuekui Yu [2,3,4,5] ✉

The portal-scaffold complex is believed to nucleate the assembly of herpesvirus procapsids. During capsid maturation, two events occur: scaffold expulsion and DNA incorporation. The portal-scaffold interaction and the conformational changes that occur to the portal during the different stages of capsid formation have yet to be elucidated structurally. Here we present high-resolution structures of the A- and B-capsids and in-situ portals of human cytomegalovirus. We show that scaffolds bind to the hydrophobic cavities formed by the dimerization and Johnson-fold domains of the major capsid proteins. We further show that 12 loop-helix-loop fragments—presumably from the scaffold domain—insert into the hydrophobic pocket of the portal crown domain. The portal also undergoes significant changes both positionally and conformationally as it accompanies DNA packaging. These findings unravel the mechanism by which the portal interacts with the scaffold to nucleate capsid assembly and further our understanding of scaffold expulsion and DNA incorporation.

Human cytomegalovirus (HCMV), the prototypical member of the *Betaherpesvirinae* subfamily, is the leading pathogenic cause of birth abnormalities, and morbidity and mortality in immunocompromised individuals, such as patients with AIDS and transplant recipients[1-5]. Recently, HCMV infection has been shown to be associated with cancer development[6].

Sharing a common architecture with other herpesviruses[7-10], HCMV consists of a lipid bilayer envelope, a pleomorphic tegument compartment and a pseudo-icosahedral nucleocapsid. The HCMV virion maturation process comprises several distinct steps, including capsid assembly, viral genome packaging, nucleocapsid tegumentation and envelopment. Capsid assembly is driven and directed by the scaffold proteins, including the assembly protein precursor (pAP) and the protease precursor (pPR), both of which can interact with the

major capsid protein (MCP) and the portal protein. The pPR and pAP auto-assemble into a sphere-like scaffold via interactions between the scaffold domains of both proteins, which, in turn, promotes MCPs and portal proteins to assemble into a fragile spherical procapsid[11-13]. The portal proteins are believed to orchestrate nucleation of the procapsid through the formation of a portal–scaffold complex, which facilitates procapsid assembly[14,15]. Although biochemical data have found several regions on scaffold and capsid proteins (i.e., MCPs and portal proteins) as being critical for capsid–scaffold interactions[16-24], the molecular details of these interactions have not been structurally elucidated.

Procapsid formation elicits the proteolytic activity of pPR. This activated pPR then cleaves both pAP and pPR, which leads to the disassociation of the scaffold from the MCP/portal. Along with cleavage of the scaffold proteins, the fragile spherical procapsid matures into a

[1]School of Pharmaceutical Science and Technology, Hangzhou Institute for Advanced Study, University of Chinese Academy of Sciences, Hangzhou 310024, China. [2]Cryo-Electron Microscopy Research Center, Chinese Academy of Sciences, Shanghai 201203, China. [3]State Key Laboratory of Drug Research, Shanghai Institute of Materia Medica, Chinese Academy of Sciences, Shanghai 201203, China. [4]University of Chinese Academy of Sciences, 100049 Beijing, China. [5]School of Chinese Materia Medica, Nanjing University of Chinese Medicine, Nanjing 210023 Jiangsu, China. [6]These authors contributed equally: Zhihai Li, Jingjing Pang. ✉e-mail: xkyu@simm.ac.cn

stable angular capsid. During the maturation process, the dis-associated scaffold proteins are expelled as the viral DNA is pumped into the capsid through the portal located at a unique vertex of the icosahedral capsid[11,12]. Three types of mature capsids—A-, B- and C-capsids—can be generated depending on the success of DNA packaging[11,25]. All three capsids possess a mature icosahedral shell and are distinguishable by the inner capsid material: (1) C-capsids are filled with viral DNA; (2) A-capsids appear empty and are assumed to be products of aborted viral genome packaging; and (3) B-capsids have a featureless inner core consisting of scaffold proteins, likely generated by an unsuccessful initiation of DNA packaging. Thus, the portals in B- and A-capsids represent to some extent the pre-DNA-packaging and the post-DNA-released states, respectively.

After being joined by several tegument proteins (i.e., the capsid vertex-specific components [CVSCs]), the nucleocapsids bud into the cytoplasm through nuclear egress. Once completely tegumented in the cytoplasm, the nucleocapsids acquire their final envelope through cytoplasmic envelopment and eventually mature into infectious virion. The B-capsid undergoes a similar maturation pathway as the nucleo-capsid and, after egressing from the host cell, becomes a noninfectious enveloped particle (NIEP)[26]. Structural analysis of the herpes simplex virus-2 (HSV-2) B-capsid at a medium resolution[27] revealed a portal with different position and conformation to that of the virion[28,29]. It has been proposed that positional and conformational changes to the portal in double-stranded DNA viruses may reflect how the portal senses the capsid inner pressure during DNA packaging[10,30,31]. Never-theless, the absence of high-resolution structural information of the portal in its potentially different functional states has limited an in-depth mechanistic understanding of the dynamic process of DNA packaging.

Here, we present the high-resolution cryo-electron microscopy (CryoEM) structures of capsids and in situ portals of nuclear A- and B-capsids, respectively. The portals in the two DNA-devoid capsids are indistinguishable, differing from the virion capsid in terms of the position and conformation of the portals. We identified the sites of the MCPs involved in the interaction with the scaffold proteins and, unexpectedly, discovered tight association between the 12 loop-helix-loop fragments of the inner scaffold protrusion and the portal crown domain. Further, we showed that only the NIEP B-capsid CVSC contains pUL48, a high molecular weight protein that is necessary for the for-mation of the genome-securing portal cap. Our results present critical structural information that facilitates an understanding of the portal–scaffold association, providing detail as to how the portal undergoes conformational changes during capsid assembly and maturation.

## Results
### CryoEM structure determination of A- and B-capsids
We purified a mixture of A- and B-capsids from the nuclei of infected MRC-5 cells, and subjected them to cryo-electron microscopy. From 7898 cryo-electron micrographs, we manually selected the A- and B-capsid particles and, from 25,502 and 31,265 particles, obtained icosahedral reconstructions of A- and B-capsids at resolutions of 3.9 and 3.7 Å, respectively (Supplementary Figs. 1 and 2 and Supplementary Table 1). The A- and B-type nuclear-derived capsids showed essentially identical capsid structures, regardless of the capsid inner components (Fig. 1a, b and Supplementary Fig. 3a–c). Through sub-particle classification and stepwise symmetry relaxation, we sequen-tially resolved the structures of the different components of the A- and B-capsids at resolutions from 4.0 Å to 7.4 Å, including the C5 portal vertex (4.1 Å for A-capsid and 4.0 Å for B-capsid), the C12 portal (4.6 Å for A-capsid and 4.2 Å for B-capsid), the C1 portal vertex (5.1 Å for A-capsid and 4.8 Å for B-capsid) and the asymmetric capsid (7.4 Å for A-capsid and 7.2 Å for B-capsid) (Fig. 1a, b and Supplementary Figs. 1 and 2). Compared with the virion capsid[30], we identified several

structural differences in the A- and B-capsids (Fig. 1c, d): (1) the portal cap, which sits atop the virion capsid portal to secure the packaged DNA from leaking out, was absent in the A- and B-capsids; (2) the pp150 tegument proteins, which decorate the virion capsid, were also absent in the A- and B-capsids; (3) the peri-portal or peri-penton CVSC helix bundle—a representative feature among herpesviruses—was absent in the A- and B-capsids; (4) the portals of the A- and B-capsids, which were devoid of DNA, were located ~20 Å inward; (5) unlike the portal turret of the virion capsid, which comprises 6 sets of upright coiled-coils, the counterpart in the A- or B-capsids contained 5 copies of coiled-coils that were inclined toward the horizontal plane.

### A hydrophobic pocket formed by dimerization and the Johnson-fold domains of MCPs mediates MCP-scaffold interaction
The A- and B-capsids possess an identical asymmetric capsid shell structure. Therefore, we determined the scaffold densities using a difference map that is produced by subtracting the asymmetric reconstruction of the A-capsid from that of the B-capsid.

The B-capsid scaffold comprises a three-layer configuration, inner core, middle band and outer shell, with tenuous densities in between (Figs. 1a and 2a, b and Supplementary Fig. 3d). The inner scaffold core is an ellipsoid of featureless densities, with radial measurements of 110 Å along the long axis and 100 Å along the short axis (Fig. 2b). The middle scaffold band exhibits a "water droplet" shape, similar to that observed in HSV-1[15], and is connected to the portal base, located between 220 and 360 Å along the long-axis radius and 200 and 340 Å along the short-axis radius (Fig. 2a, b). The outer scaffold shell, located at a radius of ~490 Å, makes extensive contacts with the inner surface of the capsid and comprises multiple icosahedrally arranged density patches, reminiscent of the arrangement observed in HSV-1[32]. Thus, we used the icosahedral reconstructions of the A- and B-capsids to gen-erate a new difference map that could provide enhanced densities of the outer scaffold shell (Supplementary Fig. 3e). From this analysis, nine density patches in the outer shell (labeled a to i) were unam-biguously identified in each asymmetric unit of the icosahedral dif-ference map (Fig. 2c, d).

Previous studies have shown that MCP–scaffold interactions rely on two hydrophobic N-terminal regions of the MCP and the C-terminal hydrophobic domain of the scaffold protein[18,24]. Indeed, after fitting the models of the capsid proteins from the HMCV virion capsid (PDB: 5vku[33]) into the icosahedral map of the B-capsid, we found that each of the 9 density patches made interactions with the capsid floor via contact with a hydrophobic pocket formed by the dimerization and the Johnson-fold domains from two neighboring MCPs (Fig. 2e, f). Thus, the density patches are most likely contributed by the C-terminal hydrophobic domains of the scaffold proteins. Given that the capsid conformational transition, including rotation and bending of the MCP floor domains, occurred primarily at the MCP molecules during capsid maturation[34], the scaffold density patches that tightly associate with MCPs—if not removed in a timely manner—would restrain the con-formational changes of the MCPs and slow down the maturation process; this hypothesis is consistent with the observation that the maturation of protease-deficient particles proceeds much slower than that of wild type; albeit both eventually reach the same endpoint[35].

### In situ portal structure of the DNA-devoid capsids
The in situ structure of the HCMV virion portal contains three sym-metry mismatches: a C5 10-helix anchor, a C6 portal turret, and a C12 portal main body[30]. The unique structural assembly of the HCMV portal was believed to be a key adaption for packaging and retention of the HCMV large genome[30]. The portals from the A- and B-capsid resemble each other and are arranged in a symmetry-mismatch fashion different from that of the virion: a C5 10-helix anchor, a C5 portal turret, and a C12 portal main body (Figs. 1c and 3a). Unlike the six-fold symmetric portal turret found in the virion capsid[30], the portal turret

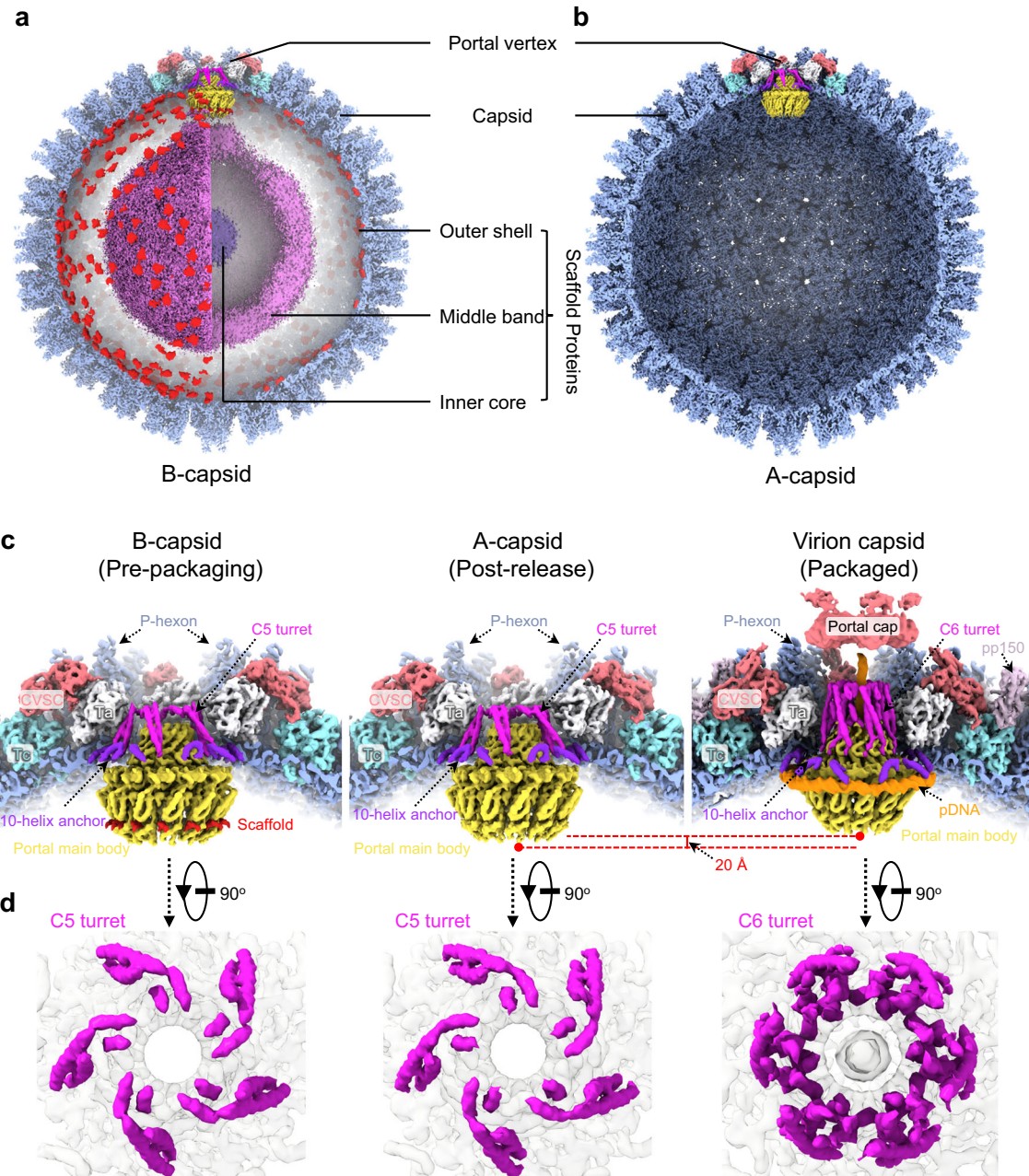

**Fig. 1 | Asymmetric reconstructions of capsids and in situ portals of the human cytomegalovirus (HCMV) A- and B-capsids.** Composite structures of the HCMV B-capsid (**a**) and A-capsid (**b**), respectively. The front-halves of both capsid shells (light blue) are removed to show their internal components. The front-right quarter of the scaffold in (**a**) is removed to show the inner three-layer scaffold (the inner core and middle band are radially colored from dark to medium purple and the outer shell is highlighted in red). The portal vertices are colored as indicated below in (**c**). **c** Close-up views of the portal vertex regions in B- (left), A- (middle) and virion (right) capsids. The capsid shell components of the portal vertex are colored by protein. The portal is colored by domain. For both B- (left) and A-capsid (middle), the composite cryoEM maps were assembled from reconstructions of C1 capsid (light blue), C5 portal vertex (CVSC, salmon; Ta, white; Tc, cyan; portal turret, magenta; portal 10-helix anchor, violet) and C12 portal main body (yellow). The portal-bound scaffold fragments from C12 reconstruction of portal in B-capsid are highlighted in red. For virion capsid (right), the composite cryoEM map were assembled from reconstructions of C1 capsid (light blue, EMD-31292), C1 portal vertex (pDNA, orange. EMD-31290), C5 portal vertex (CVSC, salmon; Ta, white; Tc, cyan; portal 10-helix anchor, violet. EMD-31297), C6 portal (portal turret: magenta. EMD-31299) and C12 portal main body (yellow. EMD-31295). **d** Top-views of in situ portals of B- (left), A- (middle), and virion (right) capsids. Portal turrets are highlighted to show the symmetry difference in arrangement between DNA-devoid and DNA-filled capsids.

resolved in DNA-devoid capsids with only five sets of coiled coils that were reclined on the C12 portal main body (Fig. 1c).

Based on the high-resolution C5 reconstruction of the portal vertices of the A- and B-capsids, we built the Cα model for the portal turret. Each coiled coil of the turret contains two long and two short helices (Fig. 3b). It is worth noting that each of the six coiled coils in the virion portal turret consists of two long and two short helices from two

neighboring portal monomers[30]. Using high-resolution C12 portal reconstruction of the A-/B-capsids, we built the atomic model for the main body part of the portal protein pUL104 (Fig. 3d, e), and showed that it consists of the wing (residues 52–90, 168–193 and 239–286), crown (residues 91–167, 194–238 and 570–647), stem (residues 287–314 and 528–551), clip (residues 315–322 and 485–527) and β-hairpin (residues 552–569) domains.

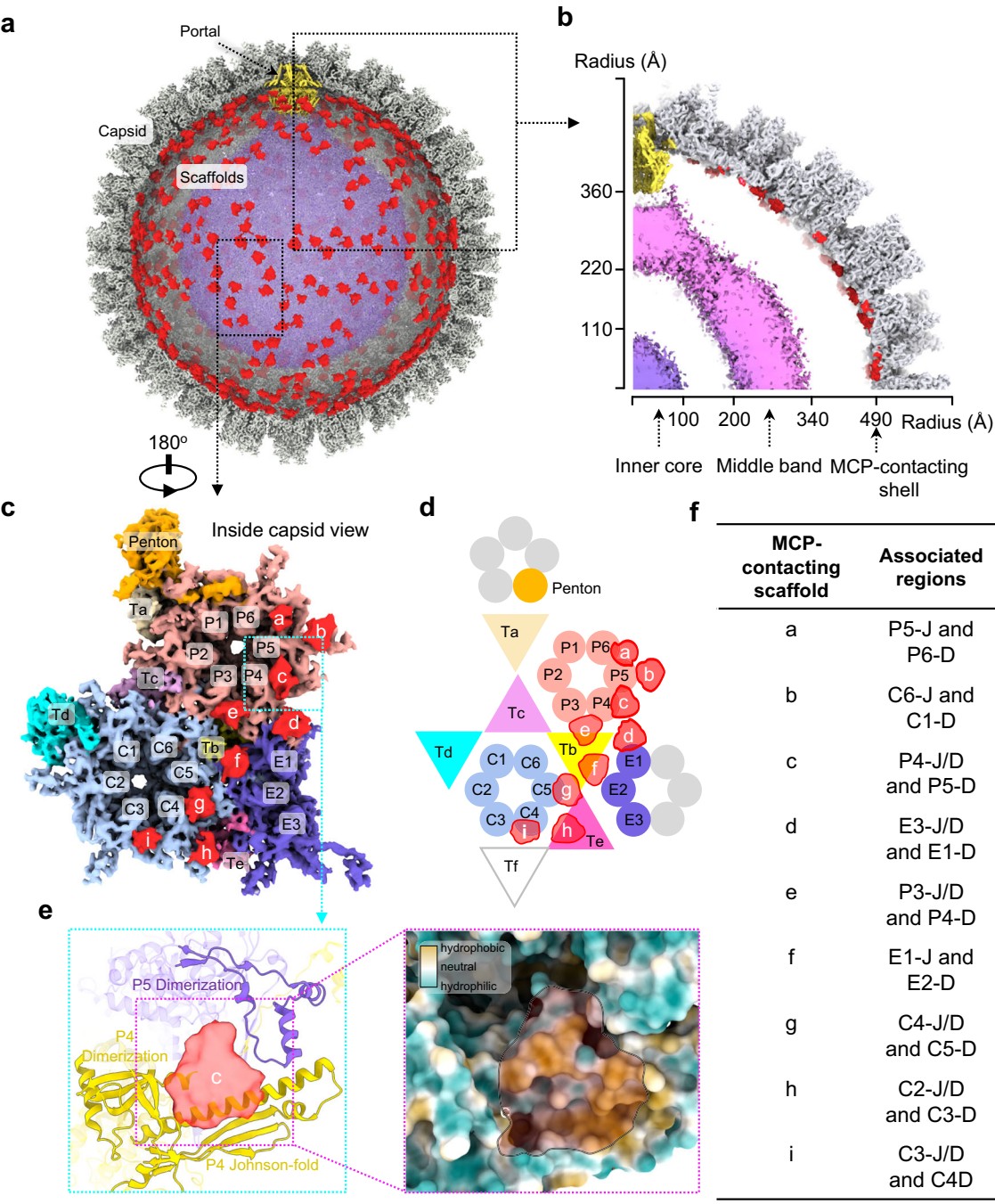

**Fig. 2 | Interactions between scaffolds and MCPs. a** Asymmetric reconstruction of the HCMV B-capsid with the front half of the capsid shell (gray) removed. The portal is in yellow and the three layers of scaffolds are colored as in Fig. 1a. **b** Cross-section of one-quarter of the B-capsid map, showing the radial distribution of the three layers of the scaffolds. Density map (**c**) and schematic (**d**) of one asymmetric unit of the icosahedral B-capsid, showing scaffold densities contacting with the capsid shell. The MCPs in center- (C-hexon), peripenton- (P-hexon), edge-hexons (E-hexon) and penton are in light blue, pink, medium purple and gold, respectively. The triplexes Ta to Te are in wheat, yellow, purple, cyan and magenta, respectively. **e** Enlarged view of the boxed region in (**c**), exemplifying that each of the nine patches of the outer shell scaffolds interacts with the capsid floor at the hydrophobic groove (inset. hydrophobic, gold; hydrophilic, cyan) formed by two neighboring MCPs. **f** Summary of the regions on MCPs associated with the nine patches of the outer shell scaffolds.

The main body of the DNA-devoid portal has a hollow central channel with three constricted regions harboring interior diameters of 30, 33, and 73 Å, respectively (Fig. 3a). Furthermore, the portal main body of the A-/B-capsid locates ~20 Å inward (Fig. 1c); in contrast, each of the 12 monomers from the portal main body of the DNA-filled capsid rotates inward. As a result, whereas the crown domain shows the greatest movement (up to 7.9 Å) toward the central channel, we noted a gradual decrease in the movements of the β-hairpin domain (with the tip sliding by 5.3 Å), wing domain (an RMSD of 2.10 Å) and stem/clip

domains (RMSDs of 1.28 Å/1.88 Å) (Fig. 3c–e). Taken together, the conformational changes to the portal in response to DNA packaging provide structural insight into the head-full mechanism of herpesvirus genome packaging.

**Interaction between scaffold and portal within the B-capsid**
Our asymmetric B-capsid reconstruction shows that the middle band of the scaffold—comprising the scaffold domain of the scaffold proteins—is associated with the base of the portal (Figs. 1a and 2b). The

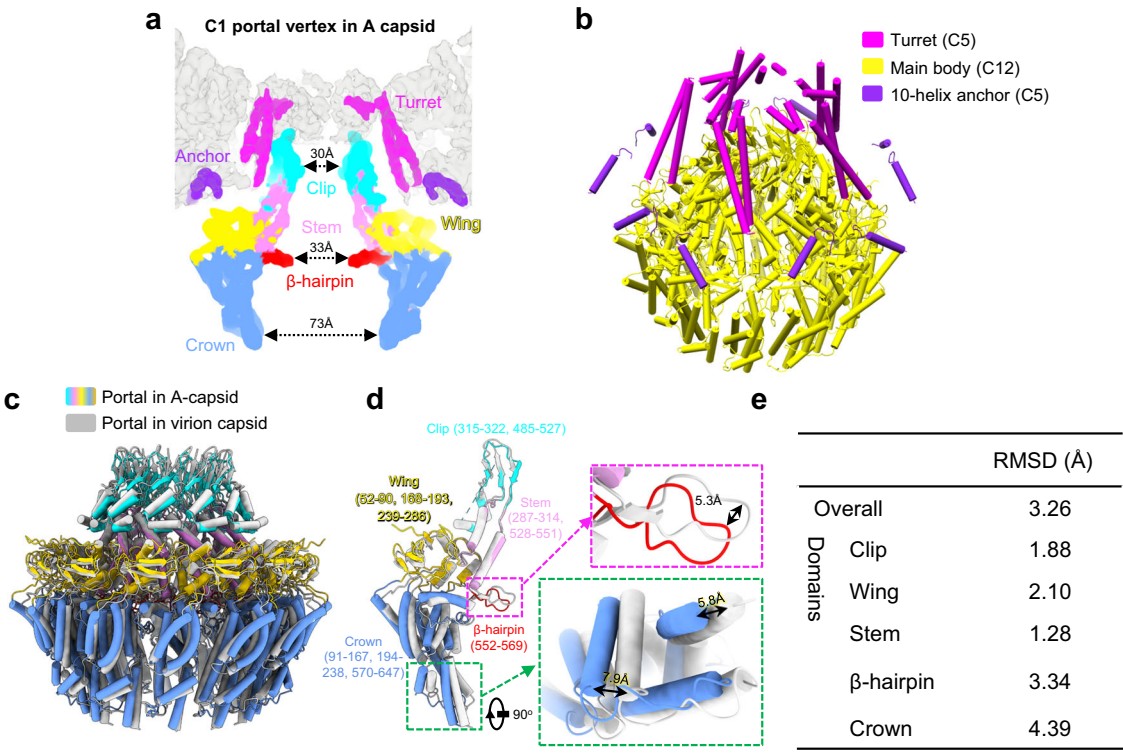

**Fig. 3 | In situ portal structure of the DNA-devoid capsid. a** Superposition of the C1 density map of the portal vertex region (transparent gray) with the segments of the C5 portal turret (magenta), C5 portal 10-helix anchor (violet), and the C12 portal main body (colored by domain: Clip, cyan; Stem, pink; Wing, yellow; β-hairpin, red; Crown, blue). **b** Pipe-and-plank depiction of the portal. The C5 turret, the C12 main body, and the C5 10-helix anchor are in magenta, yellow and violet, respectively. Structural comparison of the dodecameric (**c**) and monomeric (**d**) portal main bodies between the A-capsid (colored) and the virion capsid (gray). The portal in the A-capsid is colored by domain, as indicated in (**a**). Insets are zoomed-in views of the boxed regions in (**d**), showing significant structural differences between the A-capsid and the virion capsid. **e** Summary of the RMSDs of the different domains of the portal main body between the A-capsid and the virion capsid.

portal-associated region of the middle band, which is positionally and morphologically different from the other parts, forms a "water-droplet"-like protrusion. The regions that tightly surround this protrusion have obviously weaker and thinner densities (Figs. 1a and 2b). Notably, the C12 reconstruction of the B-capsid portal reveals that, among the cluttered scaffold densities surrounding the portal, 12 ordered density fragments tightly encircle the portal crown region (Fig. 4a and Supplementary Fig. 4). Each of the 12 ordered density fragments has comparable strength to that of the surrounding portal proteins (Supplementary Fig. 4) and well matches a 15-residue-long loop-helix-loop motif (Fig. 4a, b). More importantly, each of the scaffold fragments is accommodated into a hydrophobic cavity formed by helices from the crown domains of two neighboring pUL104 molecules (helix 110–124 from one molecule and helices 110–124, 134–164, helix-loop-helix 205–225 from another one) (Fig. 4b–d). Previous biochemical data indicated that tryptophan residues in the portal of HSV-1 are essential to this portal-scaffold interaction[23]. Consistently, we identified two tryptophan residues (Trp118 and Trp142) in the portal that participate in the interaction with the scaffold, and noted that these residues are conserved among herpesviruses (Fig. 4c and Supplementary Fig. 5).

A portal structural comparison of the DNA-free and DNA-filled capsids shows that the scaffold fragment binding cavity of the portal is closed off in the virion capsid (Supplementary Fig. 6), suggesting that dissociation of the scaffold fragments from the portal is necessary for the required portal conformational changes during DNA translocation. Given that the portal is essential for herpesviruses replication and has no counterpart in humans, the scaffold fragment binding cavity of the portal may represent a promising drug target for the development of small molecules to disrupt the incorporation of the portal into the capsid in herpesvirus infection.

## Interactions between portal and capsid proteins in DNA-devoid capsids

We docked the models of the periportal capsid proteins and the portals of the A-/B-capsids into the C1 reconstructions of the portal vertices, and identified the interactions between the portal and capsid components. The five-fold symmetrical 10-helix anchor in the DNA-devoid capsids was located beneath the floor region of the capsid, sticking to the P1 and P6 MCPs (Supplementary Fig. 7a); this is similar to that observed in the virion capsid[30].

In the DNA-filled or virion capsids, the portal has a C6 portal turret and C12 portal main body, and both parts of the portal interact with the surrounding capsid materials, which are arranged in C5 symmetry. However, in DNA-free or A/B-capsids, the portal main body remains in a C12 symmetry, with the turret of the inward-located portal arranged in a C5-fold fashion. Only these five sets of turret coiled coils interact with the surrounding capsid materials arranged in C5 symmetry (Supplementary Fig. 7a, b). Five sets of two β-hairpins from the trunk domains of one Tri1 molecule and one Tri2B molecule interact with the two long helices of the portal turret at the upper region (Supplementary Fig. 7b, c); five sets of the long helix, a short helix of P6, and two β-sheets in the floor regions of P1 and P6 make close contact with the two long helices of the portal turret at the lower regions (Supplementary Fig. 7b, d). We believe that, in addition to the 10-helix anchor, the conformation transition of the portal turret from a C-5 symmetry to a C-6 symmetry, which has only been observed in HCMV to date, would require immense inner pressure to occur, and this transition, in turn, contributes significantly to packaging HCMV DNA, the largest genome among human herpesviruses.

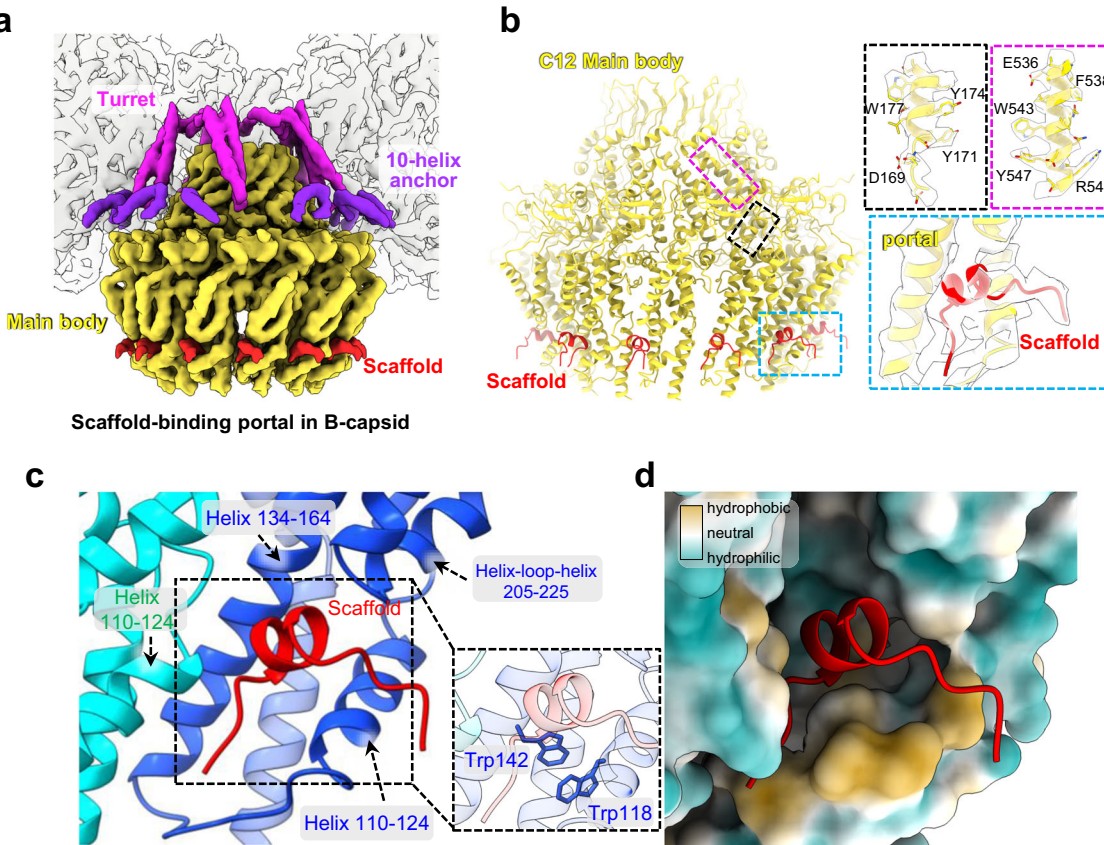

**Fig. 4 | Structure of the portal-scaffold complex in the B-capsid. a** Superposition of the C1 density map of the portal vertex region (transparent gray) with the composite cryoEM map of the portal and the portal-bound scaffold in the B-capsid that was assembled from reconstructions as in Fig. 1. **b** Atomic model of the portal main body (yellow) and the portal-bound scaffold fragments (red). Insets are zoomed-in views of the boxed regions on the left, showing the CryoEM map and model of residues 167–180 (sharpened map, contour level: 2.5σ), residues 535–550 (sharpened map, contour level: 3.0σ) of portal and the scaffold-binding region of portal-scaffold complex (unsharpened map, contour level: 2.0σ), respectively. **c, d** Zoomed-in view of the contact region between portal and one of the 12 scaffold fragments. **c** Ribbon model of the portal (colored by molecule) and the scaffold fragment (red). Inset highlights the two conserved tryptophan residues (Trp118 and Trp142) in the portal at the portal-scaffold interface. **d** Hydrophobic surface (portal, hydrophobic, gold; hydrophilic, cyan) and ribbon model (scaffold), showing the hydrophobic interaction between the scaffold fragment and the portal protein.

## pUL48 is absent for nuclear capsid CVSCs but present in NIEP-capsid CVSCs

The CVSC heteropentamer in the mature virion of HCMV consists of one pUL93, two pUL77 (the upper one and the lower one), and two pUL48 (the left one and the right one) molecules. The four-helix bundle extending to the vertex center is formed by pUL77 and pUL48, and is associated with the "base" protein, pUL93, via the N-terminal region of the lower pUL77 molecule[30]. However, neither the peri-portal nor the peri-penton CVSC of the A- or B-capsid comprises this featured four-helix bundle (Figs. 1c and 5a, b and Supplementary Fig. 8).

Based on the C1 reconstruction of the CVSC-binding penton vertex, we next built the atomic model for the B-capsid CVSC. We found that the B-capsid CVSC contains one pUL93 and two pUL77 molecules, but no pUL48 molecules (Fig. 5c–e). pUL93 in the B-capsid CVSC is essentially identical to that of the mature virion, whereas the two pUL77 proteins are only resolved in regions associated with pUL93, spanning residues 2–42 for the upper pUL77 protein, and residues 12–63 for the lower pUL77 protein (Fig. 5d, e).

Given that the nuclear B-capsid lacks the portal cap, which is formed by pUL77[30], and that the portal cap is essential to secure the packaged DNA in place, pUL48 likely offers a supportive role in portal cap formation, presumably in establishing the featured four-helix bundle with the pUL77 molecules. To explore whether the formation of the pUL48-participating helix bundle is dependent on DNA packaging, we obtained NIEPs from the supernatant of virus-infected cells and sequentially resolved the structures of the C5 portal vertex, the C12 portal, the C1 portal vertex, and the C1 capsid of the NIEP capsids at resolutions of 5, 6.9, 7.3, and 9.3 Å, respectively (Supplementary Table 1 and Supplementary Fig. 9). The asymmetric reconstruction of the NIEP capsids revealed pp150 tegument proteins that are absent in the nuclear B-capsid, as well as an inner scaffold that is similar to that of the nuclear B-capsid (Fig. 6a). The portal of the NIEP capsid is essentially identical to that of the B-capsid, consisting of a C5 10-helix anchor, a C5 portal turret and a C12 portal main body (Fig. 6b). The C12 portal main body of the NIEP capsid also comprises 12 loop-helix-loop scaffold fragments that are tightly associated with the portal crown region (Fig. 6b). Intriguingly, although no viral DNA was packaged, the NIEPs also revealed a layer of cap-like density sitting atop the portal (Fig. 6c). Nevertheless, the morphology of the portal cap in the NIEP capsid is different to that in the virion capsid (Fig. 6c). Specifically: (1) the portal cap density in the NIEP capsid has a wider diameter of ~160 Å as compared with ~125 Å in the mature virion; (2) the portal cap of the virion nucleocapsid is completely sealed, whereas the portal cap of the NIEP capsid is center-hollowed. Besides, the CVSCs of the NIEP capsids are similar to those of the virion nucleocapsids but are different to those of the nuclear B-capsids, which comprise the four-helix bundles encircling the portal vertex (Fig. 6b, c).

Overall, our results demonstrate that the formation of CVSCs can be divided into two distinct steps and that the final inclusion of pUL48

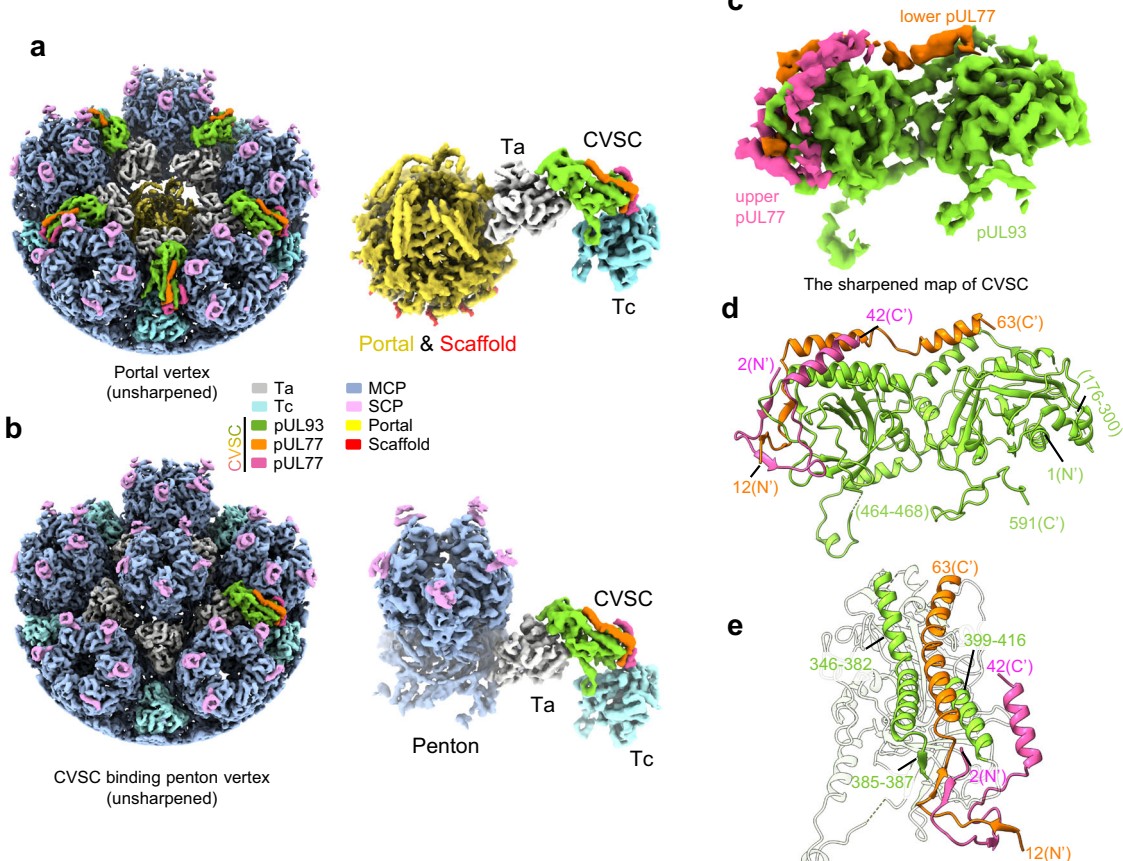

**Fig. 5 | CVSC structure in DNA-devoid capsids. a** Unsharpened maps of the portal vertex (left) and the segmentation of the portal, CVSC, Ta and Tc (right). The color codes for different molecules are indicated. **b** Unsharpened maps of the one CVSC-binding penton vertex (left) and the segmentation of the penton, CVSC, Ta and Tc (right). The color codes for different molecules are indicated. **c** Sharpened map the CVSC, colored by protein. Side (**d**) and back (**e**) views of the CVSC model. The structural motifs in (**e**) (helices 346–382 and 399–416, and strand 385–387) of the pUL93 protein that interact with pUL77 molecules are highlighted.

would automatically result in the formation of the portal cap, regardless of whether the DNA is packaged.

## Discussion

Current evidence supports that the portal interacts with scaffold proteins to nucleate the assembly of procapsids[12–15]. Although one biochemical study identified a region in the HSV-1 scaffold protein that is required for its interaction with the portal[24], it remains unclear how the portal interacts with the scaffold proteins. The present study shows that the portal of the B-capsid makes extensive associations with a protruded region of the middle band of the scaffold, which comprises the scaffold domain (Figs. 1a and 2b) and 12 loop-helix-loop scaffold fragments that insert into the pockets of the portal crown domain (Figs. 1 and 4). In addition, the portal of the B-capsid strongly interacts with the surrounding capsid proteins in a C5 symmetrical manner (Supplementary Fig. 7). Given that B-capsids are thought to mature without encountering the DNA encapsulation machinery[25,36] and the B-capsid portal somewhat represents the pre-DNA-packaging state, the portal-scaffold protrusion complex revealed in the B-capsid may explain how the portal interacts with the scaffold proteins to initiate procapsid assembly.

Maturation of the procapsid to become a DNA-encapsulated C-capsid is accompanied by expulsion of the scaffold. A previous study has shown that inhibition of genome-terminase complex formation generates only B-capsids[37]; thus, it is likely that scaffold elimination and DNA packaging are coupled and coordinated[38]. While the proteolytic cleavage process disassociates the scaffold from the MCP, it is

likely that the initialization of DNA packaging triggers the disengagement of the portal from the scaffold, which may tear open the scaffold from the "water-drop" protrusion region to facilitate its expulsion, thereby making room for the viral genome. Meanwhile, disengagement of the bound scaffold fragments from the portal would, in turn, release the restraint on the portal to allow for further conformational changes that are required for DNA packaging to proceed. Given that the portal in the A-capsid, which somewhat represents the post-release state, is identical to that in the B-capsid both conformationally and positionally, we believe that inner capsid pressure generated by the packaged DNA causes the conformational changes of the portal.

## Methods

### HCMV capsid preparation

Human fibroblast MRC-5 cells (ATCC, #CCL-171) were grown in Eagle's Minimal Essential Medium (EMEM) supplemented with 10% fetal bovine serum (FBS), and were incubated in a humidified incubator with 5% $CO_2$ at 37 °C. At ~90% confluence, cells were infected with the HCMV strain AD169 at a multiplicity of infection (MOI) of 2, and then incubated in a humidified incubator with 5% $CO_2$ at 37 °C. Five hours after infection, the culture medium was replaced with fresh medium containing 10% FBS. At 4 days' post-infection, when roughly 80% of the cells had become swollen, the adherent cells were collected by scraping and pelleted via centrifugation at $1000 \times g$ at 4 °C for 10 min. The pellets were washed with PBS, resuspended in PBS containing 1% NP40 solution, and then incubated on ice for 5 min. The cell nuclei were pelleted by centrifugation at $3000 \times g$ at 4 °C for 5 min, and then

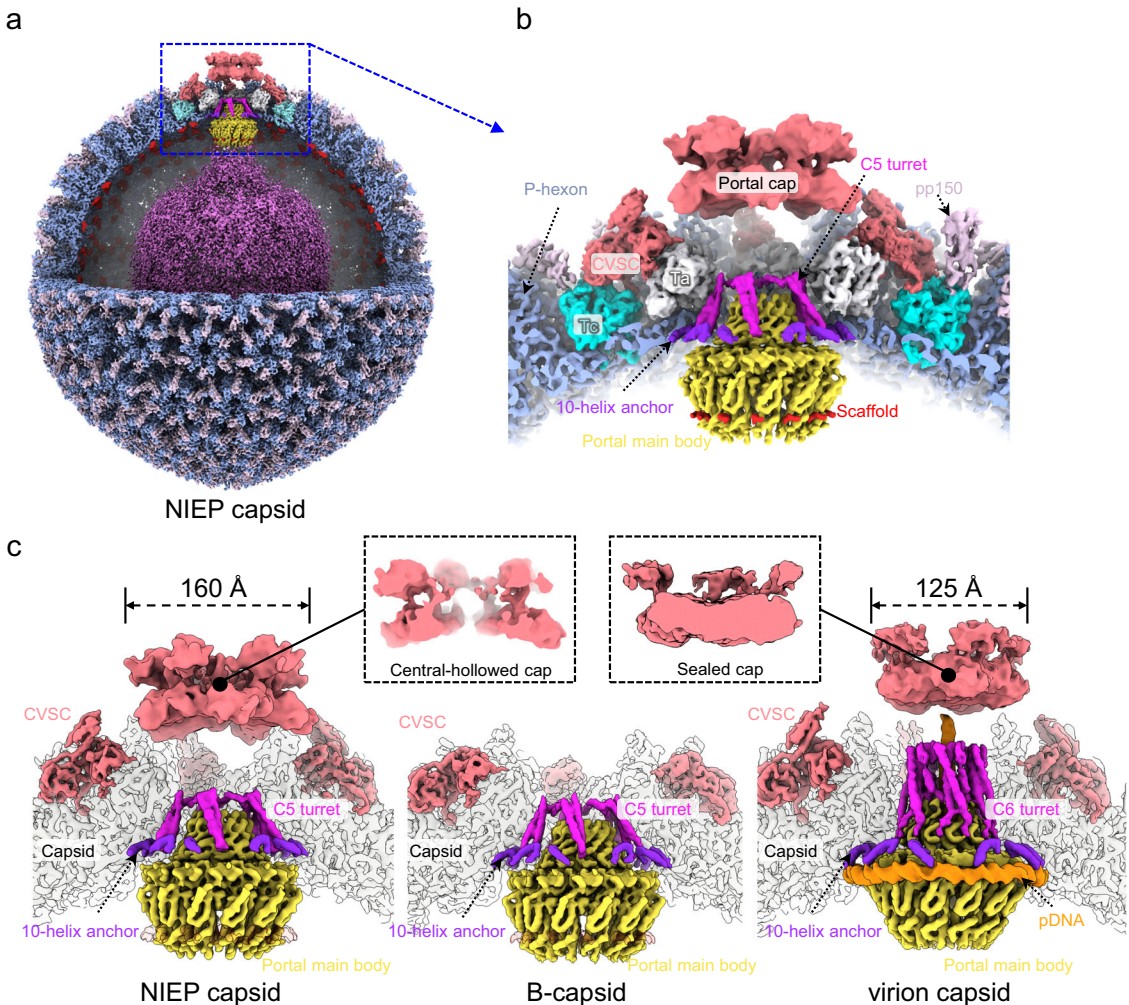

**Fig. 6 | Asymmetric reconstructions of capsids and in situ portal of NIEP capsids. a** Composite map of the asymmetric structure of the NIEP capsid. Upper-front quarter of the capsid and the outer scaffold shell are removed to show the portal vertex and inner capsid materials. The composite map is assembled from reconstructions of C1 capsid (capsid, light blue; pp150, pink), C5 portal vertex (CVSC, salmon; Ta, white; Tc, cyan; portal turret, magenta and 10-helix anchor of portal, violet) and C12 portal (portal main body, yellow and scaffold, red). **b** Zoomed-in view of the boxed region in (**a**), showing the structural architecture of the portal vertex in the NIEP capsid. **c** Structural comparison of the portal vertex between NIEP- (left), B- (middle) and virion capsid (right). Insets, cross-sections of the portal cap of the NIEP- and virion capsid, respectively. The composite maps of the B- and virion capsids are assembled using the same components as in Fig. 1.

resuspended in 2 ml of lysis buffer (500 mM NaCl, 10 mM Tris, 1 mM EDTA, pH 7.5) containing 0.1% NP40, 100 µl DNase I (1 U/µl) and 100 µl DNase I reaction buffer. To degrade the nuclear membrane and release the capsid, the nuclear resuspension was passed 20 times through a 23-gauge hypodermic needle. The solution was centrifuging at 2000 × $g$ for 5 min, and the supernatant collected and sedimented through a 40% (w/v) sucrose solution (containing 0.1% NP40) at 112,400 × $g$ for 1 h at 4 °C. The pellet was resuspended in lysis buffer containing 0.1% NP40 and then further purified by centrifugation through a 22 to 50% (w/v) continuous sucrose gradient at 70,000 × $g$ for 1 h. The two light-scattering bands containing the A-capsid and B-capsid, respectively, were collected and merged. The merged solution was diluted with lysis buffer containing 0.1% NP40 to a volume of 13 ml and the capsids pelleted by centrifugation at 90,000 × $g$ for 1 h. The pellet was finally resuspended in 10 µl of lysis buffer containing 0.1% NP40 for further cryoEM experiments.

## HCMV NIEPs preparation

MRC-5 cells were grown in normal media to ~80% confluence and then infected with HCMV at MOI = 0.5. Twelve hours after infection, the culture medium was replaced with fresh medium containing 2% FBS and the cells incubated in a humidified incubator with 5% $CO_2$ at 37 °C.

At 14 days' post-infection, when ~80% of cells had lysed, the culture media was collected and centrifuged at 10,000 × $g$ for 12 min to remove cell debris. The supernatant was collected and centrifuged at 80,000 × $g$ for 1 h to pellet the particles. The pellet was then resuspended in PBS and further purified by centrifugation through a 15 to 50% (w/v) continuous sucrose gradient at 70,000 × $g$ for 1 h. Two light-scattering bands were observed; the upper band containing the NIEP particles was collected, diluted with PBS to a volume of 13 ml, and then pelleted by centrifugation at 90,000 × $g$ for 1 h. The pellets were finally resuspended in 10 µl of PBS.

## CryoEM sample preparation and data collection

CryoEM grids of the HCMV nuclear-derived capsids were prepared by applying 2.5-µl aliquots of sample to a glow-discharged 300-mesh Quantifoil grid (R1.2/1.3). The grids were blotted with filter paper for 14.0 s and frozen by plunging into liquid ethane using an FEI Vitrobot IX.

To prepare cryoEM samples of the NIEP capsids, the viral samples were mixed with Triton X-100 to a final concentration of 1.2% immediately before preparation of the cryoEM grids. These NIEPs samples were then frozen using the same process described above for nuclear-derived capsid samples.

CryoEM movies for both samples of nuclear-derived and NIEPs capsids were collected on a Titan Krios microscope (FEI) equipped with a Gatan Imaging filter (GIF) and a K3 direct electron detector in super-resolution mode. The microscope was operated at 300 kV with a nominal magnification of ×53,000, yielding a calibrated pixel size of 0.8125 Å for the specimen. Using the software package SerialEM[39], a total of 7879 and 10,124 movies were collected for the capsid and the NIEPs samples, respectively, at a dose rate of 10 electrons/Å$^2$ s for 3 s.

### CryoEM image processing and icosahedral reconstruction of HCMV A- and B-capsid

For each movie stack, the 30 frames were aligned by beam-induced motion correction with the program MotionCor2[40], and dose-weighted frames with 2 time-binning in each stack were used for further processing. The defocus values and astigmatism parameters for each micrograph were determined by CTFFIND4[41]. In the micrographs, the B-capsid proteinaceous core allows for easy differentiation from the empty A-capsid. In total, 33,110 A-capsids and 33,480 B-capsids were separately and manually picked using the Manual Picking function in Relion 3.0[42]. To accelerate image processing, the boxed particle images were binned eight times (256 × 256) before 2D and 3D classifications. A 30-Å low-pass filter map of the HCMV virion capsid (EMDB-8703) was used as a reference for 3D classifications. In the end, a total of 25,502 and 31,265 particles for A- and B-capsids, respectively, from good 3D classes were selected and re-extracted with 2 time-binning (1024 × 1024) for 3D refinement. CryoEM structures of the A- and B-capsids at resolutions of 4.5 and 4.3 Å, respectively, were obtained by conventional icosahedral reconstruction in Relion 3.0. After correcting the Ewald-sphere curvature by adding the argument −*Ewald* in the *relion_reconstruction* for both half maps, we finally improved the icosahedral reconstructions of the A- and B-capsids to resolutions of 3.9 and 3.7 Å, respectively.

### Structure determination of the portal vertex, one CVSC binding penton vertex, portal dodecamer, and asymmetric reconstruction of the A- and B-capsids

Structure determination of the non-icosahedral symmetric elements of the A- and B-capsids—the portal vertex, portal, virion capsid, and one-CVSC-binding penton vertex—are illustrated in Supplementary Fig. 1. In brief, with the aid of the icosahedral orientations and center parameters determined above, the 12 vertex sub-particles of the A- and B-capsids, respectively, were located and extracted with a Scipion plugin Localized_Reconstruction[43]. The refined vertex sub-particles were then subjected to a round of focus alignment (C5). One of the six converged classes, responsible for 7.2% and 7.0% of the sub-particle datasets for the A- and B-capsids, respectively, showed prominent portal features beneath the capsid floor, whereas all the other classes displayed a penton at the center. Keeping the one with the highest score in the flag _rlnMaxValueProbDistribution in *particle.star* file, we removed the redundant sub-particles to yield portal vertex datasets for the A- and B-capsids of 22,059 and 26,181 sub-particles, respectively. Finally, we obtained C5 reconstructions of the portal vertices of the A- and B-capsids at resolutions of 4.1 and 4.0 Å, respectively.

To resolve the structures of the portal main body, we expanded the dataset of the portal vertices of the A- and B-capsids with five-fold symmetry, respectively, further extracted the sub-particles that covered only the central lower part of the portal vertex, and then performed a round of 3D classification (C12) with rotation alignment disabled (--*skip_align*). Among the 6 classes generated, we selected the one that showed a dodecameric portal structure with a ratio closest to 20.0%, and removed particle redundancy in that class. Finally, we obtained a total of 14,039 and 18,426 portal sub-particles for the A- and B-capsids, respectively, and used these results to determine the 12-fold symmetric reconstructions of the portal at resolutions of 4.6 and 4.2 Å for the A- and B-capsids, respectively. Based on the orientation

parameters (_rlnAngleRot, _rlnAnglePsi and _rlnAngleTilt) of the portal sub-particles, the corresponding portal vertex sub-particles and capsid particles were used to reconstruct the asymmetric structures of the portal vertex and the capsid. Finally, we obtained the C1 reconstructions of the portal vertex and the capsid at global resolutions for the A-capsid of 5.1 and 7.4 Å, respectively, and for the B-capsid of 4.8 and 7.2 Å, respectively.

The structure of the one-CVSC-binding penton vertex of the B-capsid was determined as previously described[10,30]. Briefly, we used five-fold symmetry to expand the dataset of the penton vertex sub-particles generated from the portal vertex-isolated 3D classification above to perform a round of 3D classification without rotation searching (--*tau = 40*) with a mask covering only one CVSC. Among the converged 6 classes, the one (14.8%) showing comparable CVSC densities with its surrounding capsid proteins was selected. In the generated CVSC-binding dataset, we selected the penton vertex sub-particles with only one CVSC and finally obtained reconstruction of the one-CVSC-binding penton vertex (C1) at a resolution of 4.1 Å from 40,903 sub-particles.

The global and local resolutions for all reconstructions were determined by gold-standard Fourier shell correlation using the 0.143 threshold[44] and ResMap[45], respectively.

### CryoEM image processing of the NIEPs capsid

Icosahedral reconstruction and sub-particle refinement of the NIEPs capsids were performed as above described. The final maps of the C5 portal vertex, C12 portal, C1 portal vertex and C1 capsid were determined at global resolutions of 5.0 Å (10,825 particles), 6.9 Å (8631 particles), 7.3 Å (8631 particles) and 9.3 Å (8631 particles), respectively.

### Model building

To build atomic models of the portal main body of the nuclear-derived A- and B-capsids, the portal dodecameric model of the virion capsid was fitted into the C12 reconstructions of the portals for the A- and B-capsids, respectively. The models of the portal were then manually adjusted in COOT[46]. Because we are unable to determine the sequence based on the C12 density map of the B-capsid portal, we only traced and built a Cα model of 15-residues: long loop-helix-loop motif for the portal-bound scaffold. Finally, the models of the portal of the A-capsid and the portal-scaffold complex of the B-capsid were subjected to refinement with PHENIX[47] against the C12 portal reconstructions of the A- and B-capsids, respectively. Owing to the relatively lower local resolution for the portal turret region in the C5 map of B-capsid portal vertex, we built the Cα model for each coiled coil into these densities, which contains two long and two short helices. Figures were prepared using ChimeraX[48].

### Reporting summary

Further information on research design is available in the Nature Portfolio Reporting Summary linked to this article.

## Data availability

All density maps generated in this study have been deposited in the Electron Microscopy Bank under accession codes EMD-34698 (Icosahedral A-capsid), EMD-34691 (C5 portal vertex of A-capsid), EMD-34692 (C12 portal of A-capsid), EMD-34694 (C1 portal vertex of A-capsid), EMD-34695 (C1 A-capsid), EMD-34699 (Icosahedral B-capsid), EMD-34696 (C5 portal vertex of B-capsid), EMD-34693 (C12 portal of B-capsid), EMD-34697 (C1 portal vertex of B-capsid), EMD-34704 (C1 CVSC-binding penton vertex of B-capsid), EMD-34700 (C1 B-capsid), EMD-34701 (C5 portal vertex of NIEPs capsid), EMD-34702 (C12 portal of NIEPs capsid), EMD-34703 (C1 portal vertex of NIEPs capsid) and EMD-34706 (C1 NIEPs capsid). The atomic coordinates generated in this study have been deposited in the Protein Data Bank under accession code 8HEU (C12 portal of A-capsid), 8HEV (C12 portal

binding with scaffold fragments of B-capsid), 8HEX (C5 portal vertex of B-capsid) and 8HEY (C1 one CVSC-binding penton vertex), respectively. The density maps of HCMV virion capsid used in this study are available in the Electron Microscopy Bank under accession codes EMD-31292 (C1 capsid), EMD-31290 (C1 portal vertex), EMD-31297 (C5 portal vertex), EMD-31299 (C6 portal) and EMD-31295 (C12 portal main body).

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

## Acknowledgements

The cryo-EM data were collected at Cryo-Electron Microscopy Research Center, Shanghai Institute of Material Medica. This work was partially supported by the National Key Research and Development Program of China (2022YFC3400500 to X.Y. and 2022YFC2804800 to Z.L.); the 100 Talents Program of the Chinese Academy of Sciences (to X.Y.); the Lin-gang Laboratory (LG202103-02-08); the Shanghai Municipal Science and Technology Major Project (TZX022021007 to X.Y.); Zhejiang Provincial Natural Science Foundation of China (LY23H190002 to Z.L.); the Research Funds of Hangzhou Institute for Advanced Study (2022ZZ01010 to Z.L.).

## Author contributions

X.Y. conceived the project, designed and supervised the research. J.P. and M.Z. prepared samples. Z.L., J.P. and Q.W. collected the data. Z.L., J.P. and R.G. determined the structures. Z.L. and J.P. built the models and prepared figures. X.Y. and Z.L. interpreted the results and wrote the manuscript.

## Competing interests

The authors declare no competing interests.
