## [Peer Review File · Nature Communications]

REVIEWER COMMENTS

Reviewer #1 (Remarks to the Author):

The authors present three cryoEM structures of various forms of the HCMV capsid at good resolutions to facilitate atomic level interpretation. The similarity between the A and B capsids is expected and are nicely illustrated. The location of the portal and its interaction with the scaffold in the B-capsid is particularly noted. The structure of the NIEPs as a failed virion is further interesting. Perhaps the biggest omission is not adequately addressing the apparent C5 to C6 transition in the portal turret. Because these lie on the interface with symmetry mismatch, could it be an artifact of whether C5 or C6 symmetry is imposed?

I find the legends not comprehensive enough to understand the figures on their own. Necessary details about the color assignments and labeling are missing. Both the main body figures and the extended data figures need to be properly narrated to be understandable.

There are several statements that are related to the content of the following missing reference: Buch et al. 2021, 12(2), e03575-20

Lines 54 & 282:

There is evidence that herpes capsids can form with no portal, as well as with more than one portal (e.g. see Buch et al.).

Line 130:

The scaffold spans the radius inside the capsid. Something should be mentioned about connections between the three layers.

Line 134:

The water droplet shape was also observed in Buch et al.

Line 236:

How does a C5 arrangement switch to a C6 arrangement? What evidence is there that it only occurs in HCMV?

Line 303:

A much larger outward movement of the portal during maturation of the procapsid has been noted in Buch et al.

Line 305:

"The portal conformational changes—particularly the outward movement presumably caused by DNA packaging—would not only promote disassociation of the scaffold from the portal but may also directly tear open the scaffold to facilitate its expulsion, thereby making room for the viral genome".

This sentence has a causal logical problem: If the DNA packaging causes outward movement of the portal, that then subsequently tears the scaffold, that then makes room for DNA packaging, how does the DNA packaging start in the first place? It is more likely that the cleavage of the scaffold by the protease releases it from both the major capsid protein and the portal, making room for the initial packaging of the DNA. The subsequent concurrent DNA packaging and scaffold expulsion then leads to the final outward movement and conformational changes of the portal.

Figure 1:

The color coding needs to be given in the legend. What are the red densities in panel a? Where is the virion reconstruction in panel c from?

Line 341:

Why were the cells lysed for purifying NIEPs? The NIEPs are supposed to be extracellular and in the original preparation by Irmiere and Gibson they did not break the cells. Also, line 259 states that the NIEPs were obtained from the supernatant. This really needs to be clarified.

Reviewer #2 (Remarks to the Author):

The assembly of herpesvirus capsids involves the formation of a procapsid, expulsion of the scaffold and incorporation of DNA. During this process two abortive particles are generated: B-capsids retain the scaffold; and A-capsids have expelled the scaffold but have not incorporated the DNA. B-capsids can also continue the virion assembly route, generating non-infectious enveloped particles (NIEPs).

In this manuscript, Li et al study human cytomegalovirus (HCMV) B- & A-capsids and capsids from NIEPs, focusing on the portal. They unravel important differences between the B- & A-capsid portals, and the virion capsid portal, providing insights into the HCMV capsid maturation process. They also identify interactions between the scaffold and capsid and the scaffold and the portal (in B-capsids), and compare the portals from B-capsids and NEIP capsids. Overall, the manuscript is well written, the results and conclusions are sound, and the cryo-EM averages & the resulting atomic models are impressive. I believe this article will gain significant interest from the structural biology and virology fields.

Minor comments:

- The authors specify in the introduction (line 74) that the portals in the B- and A-capsids somewhat represent the pre-DNA-packaging and the post-DNA-released states. They should also repeat this in the discussion, as one of the limitations of their study.
- The authors should explain in more detail how the turret can change its conformation during the viral life cycle (i.e. the fact that the turret has a C5 symmetry in B- and A-capsids, while it has C6 symmetry in virion capsids). This would mean that the turret transitions between different symmetries during assembly and infection, which is not straightforward to explain (i.e. it would assemble with C5 symmetry, then transition to C6 when the DNA is incorporated, and then go back to C5 after the DNA is being expelled). The authors mention this transition could be a consequence of the DNA pressure, but it is hard to reconcile this with the fact that other herpesvirus turrets are C5, even under the pressure of the encapsidated DNA.
- Line 115: I assume that the authors are using EMDB-8703 for the virion capsid EM average; this should be mentioned in the text.
- Fig 2e: I might be misinterpreting this, but it seems that the hydrophobic scale bar is inverted. The outer scaffold shell density seems to be interacting with a large patch of capsid floor that is hydrophilic (orange).
- Extended Data Figure 2: the authors need to rearrange the labels and/or update the legend, as the two do not match (e.g. there is no panel d).

We would like to express our gratitude to the editor and the two reviewers for spending time in evaluating our paper. As you will see from our point-by-point response below, we have been thorough in our attempts to address all the points raised by the reviewers. We have revised the paper accordingly.

To facilitate your navigation of this response statement, the referees' comments are in **black**, and our responses are in **blue**.

Referees' comments:

Reviewer #1 (Remarks to the Author):

The authors present three cryoEM structures of various forms of the HCMV capsid at good resolutions to facilitate atomic level interpretation. The similarity between the A and B capsids is expected and are nicely illustrated. The location of the portal and its interaction with the scaffold in the B-capsid is particularly noted. The structure of the NIEPs as a failed virion is further interesting. Perhaps the biggest omission is not adequately addressing the apparent C5 to C6 transition in the portal turret. Because these lie on the interface with symmetry mismatch, could it be an artifact of whether C5 or C6 symmetry is imposed?

Re: We thank the reviewer for the positive comment on our work. As for the structures of the portal turrets of nucleic capsid (A-/B-capsid) and virion capsid in HCMV, we identified their symmetric properties solely based on the C1 reconstructions of portal vertices in those capsids (See Supplementary Fig. 5 in Li, et al, Nat Commun 2021; and Fig. 2a in our revised manuscript). Then, we imposed C5 or C6 symmetry averaging onto the portal vertex reconstruction to improve the density quality of the portal turret. Therefore, the C5 or C6 symmetry of the portal turret is not an artifact of different symmetry imposed.

I find the legends not comprehensive enough to understand the figures on their own. Necessary details about the color assignments and labeling are missing. Both the main body figures and the extended data figures need to be properly narrated to be understandable.

Re: Following the suggestion of the reviewer, we have significantly modified the figure legends in our revised manuscript.

There are several statements that are related to the content of the following missing reference: Buch et al. 2021, 12(2), e03575-20

1. Lines 54 & 282:

There is evidence that herpes capsids can form with no portal, as well as with more than one portal (e.g. see Buch et al.).

Re: It is correct that a previous study using the cryoET method has shown that a very small proportion of capsids have no or two portals. We are sorry for missing this reference. We have revised those statements accordingly and included the reference in the revised manuscript.

2. Line 130:

The scaffold spans the radius inside the capsid. Something should be mentioned about connections between the three layers.

Re: Following the reviewer's suggestion, we have added a figure (updated Extended Data Fig. 3d) to show the tenuous densities among the three scaffold layers and revised the manuscript accordingly.

3. Line 134:

The water droplet shape was also observed in Buch et al.

Re: We thank the reviewer for pointing this out. We have included this fact in the revised manuscript.

4. Line 236:

How does a C5 arrangement switch to a C6 arrangement? What evidence is there that it only occurs in HCMV?

Re: It would be significant to reveal the dynamic process associated with conformation switching of the portal turret in HCMV as DNA-packaging proceeds. However, due to a lack of high-resolution structure information for the in-situ portal

from the capsids at the different transition states along the dynamic trajectory, our knowledge of this dynamic process from C5 to C6 arrangement is still limited at this stage.

We do not intend to say that the C5-to-C6 switch only occurs in HCMV, but that this switch has only been observed in HCMV to date. As among the reported structures of in-situ portal in herpesviruses (McElwee, et al, PLoS Biol 2018; Liu, et al, Nature 2019; Gong, et al, Cell 2019; Li, et al, Cell Research 2020; Wang, et al, Protein & Cell 2020), only HCMV was identified as having a portal turret with a C5-to-C6 transition during DNA packaging. We have changed our statement “only observed in HCMV” to “which has only been observed in HCMV to date,” in the revised manuscript to avoid any confusion.

5. Line 303:

A much larger outward movement of the portal during maturation of the procapsid has been noted in Buch et al.

Re: We thank the reviewer for pointing this out. We have included this reference in the revised manuscript.

6. Line 305:

"The portal conformational changes—particularly the outward movement presumably caused by DNA packaging—would not only promote disassociation of the scaffold from the portal but may also directly tear open the scaffold to facilitate its expulsion, thereby making room for the viral genome".

This sentence has a causal logical problem: If the DNA packaging causes outward movement of the portal, that then subsequently tears the scaffold, that then makes room for DNA packaging, how does the DNA packaging start in the first place? It is more likely that the cleavage of the scaffold by the protease releases it from both the major capsid protein and the portal, making room for the initial packaging of the DNA. The subsequent concurrent DNA packaging and scaffold expulsion then leads to the final outward movement and conformational changes of the portal.

Re: We thank the reviewer for indicating the logical problem with this sentence. We have revised the corresponding text in our updated manuscript according to the reviewer's suggestion.

7. Figure 1:

The color coding needs to be given in the legend. What are the red densities in panel a? Where is the virion reconstruction in panel c from?

Re: We are sorry for the confusion. The red densities in panel a reflect the outer shell of the scaffold. The virion reconstruction in panel c is a composite reconstruction from our previously published paper (Li, et al, Nat Commun 2021). In our revised manuscript, we now include the color code within the legends. We also indicate the accession IDs of the reconstructions associated with the assembly of the composite map of the virion capsid in panel c.

8. Line 341:

Why were the cells lysed for purifying NIEPs? The NIEPs are supposed to be extracellular and in the original preparation by Irmiere and Gibson they did not break the cells. Also, line 259 states that the NIEPs were obtained from the supernatant. This really needs to be clarified.

Re: We did not break the cells for NIEP preparation. We collected the extracellular culture media after the host cells were lysed naturally by the infecting virus, which led to the collection of a high concentration of NIEPs. We apologize for the confusion and have modified the text in Methods section in the revised manuscript.

Reviewer #2 (Remarks to the Author):

The assembly of herpesvirus capsids involves the formation of a procapsid, expulsion of the scaffold and incorporation of DNA. During this process two abortive particles are generated: B-capsids retain the scaffold; and A-capsids have expelled the scaffold but have not incorporated the DNA. B-capsids can also continue the virion assembly route, generating non-infectious enveloped particles (NIEPs).

In this manuscript, Li et al study human cytomegalovirus (HCMV) B- & A-capsids and capsids from NIEPs, focusing on the portal. They unravel important differences between the B- & A-capsid portals, and the virion capsid portal, providing insights into the HCMV capsid maturation process. They also identify interactions between the scaffold and capsid and the scaffold and the portal (in B-capsids), and compare the portals from B-capsids and NEIP capsids. Overall, the manuscript is well written, the results and conclusions are sound, and the cryo-EM averages & the resulting atomic models are impressive. I believe this article will gain significant interest from the structural biology and virology fields.

Re: We are very grateful for the reviewer's positive comment on our study.

Minor comments:

1. The authors specify in the introduction (line 74) that the portals in the B- and A-capsids somewhat represent the pre-DNA-packaging and the post-DNA-released states. They should also repeat this in the discussion, as one of the limitations of their study.

Re: Following the reviewer's suggestion, we have provided further discussion about the B- and A-capsids in the revised manuscript.

2. The authors should explain in more detail how the turret can change its conformation during the viral life cycle (i.e. the fact that the turret has a C5 symmetry in B- and A-capsids, while it has C6 symmetry in virion capsids). This would mean that the turret transitions between different symmetries during assembly and infection, which is not straightforward to explain (i.e. it would assemble with C5 symmetry, then transition to C6 when the DNA is incorporated, and then go back to C5 after the DNA is being expelled). The authors mention this transition could be a consequence of the DNA pressure, but it is hard to reconcile this with the fact that other herpesvirus turrets are C5, even under the pressure of the encapsidated DNA.

Re: We thank the reviewer for this insightful comment and we agree that it is difficult to explain how the turret changes its conformation during the viral life cycle. We believe that the C5-to-C6 transition of the portal turret is an intrinsic property of the HCMV portal. Furthermore, we surmise that a much larger inner capsid pressure in HCMV, as compared with other herpesviruses, is required to make this transition happen, which, in turn, facilitates the packaging of the large HCMV DNA.

3. Line 115: I assume that the authors are using EMD8-8703 for the virion capsid EM average; this should be mentioned in the text.

Re: The map of the virion capsid used in Figure 1 is a composite structure from several individual reconstructions. We have included the accession ID of each reconstruction in the revised legend for Figure 1.

4. Fig 2e: I might be misinterpreting this, but it seems that the hydrophobic scale bar is inverted. The outer scaffold shell density seems to be interacting with a large patch of capsid floor that is hydrophilic (orange).

Re: We are sorry for the typographical error; this has been corrected in the revised manuscript.

5. Extended Data Figure 2: the authors need to rearrange the labels and/or update the legend, as the two do not match (e.g. there is no panel d).

Re: This error has been corrected.